# HIGHLIGHT DIFFUSION: TRAINING-FREE ATTENTION GUIDED ACCELERATION FOR TEXT-TO-IMAGE MODELS

## ABSTRACT

Diffusion models have achieved exceptional results in image synthesis, yet their sequential processing nature imposes significant computational demands and latency, posing challenges for practical deployment. In this paper, we present Highlight Diffusion: a training-free novel acceleration approach that achieves noticable speedup while retaining generation quality through an attention-guided generation process. By utilizing cross-attention maps to identify crucial segments within the image, we selectively compute these highlighted regions during the denoising process, bypassing the need for full-resolution computation at every step. This strategy maintains high-quality outputs while enabling faster, more resource-efficient diffusion model inference. With minimal loss in generated image quality—evidenced by only a 0.65 increase in FID score and a 0.02 decrease in CLIP score, Highlight Diffusion achieved a 1.52× speedup using an NVIDIA RTX 3090 GPU.

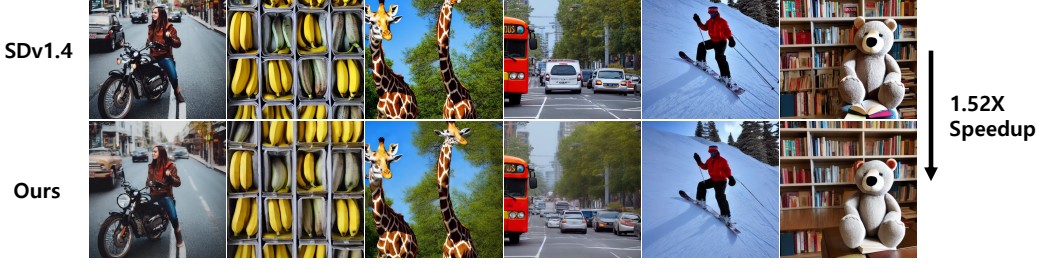

SDv1.4

Ours

1.52X
Speedup

Figure 1: Comparison of the produced Image of Stable Diffusion V1.4 (Top) and Highlight Diffusion (Bottom)

## 1 INTRODUCTION

Diffusion models (Ho et al., 2020; Song et al., 2020; Dhariwal & Nichol, 2021; Rombach et al., 2022) have recently emerged as a highly effective approach to image synthesis, setting new benchmarks across various tasks. By iteratively transforming random noise into structured data, models such as **Denoising Diffusion Probabilistic Models (DDPM)** have demonstrated an exceptional ability to generate high-quality images with impressive fidelity and diversity compared to the GANs (Goodfellow et al., 2020) and VAEs (Kingma, 2013). Despite these advancements, practical deployment of diffusion models remains constrained by their computational complexity and high latency, which are byproducts of their large model sizes and inherently sequential nature.

To mitigate these challenges, recent research has focused on accelerating diffusion models through various techniques, such as reducing the number of diffusion steps through ODE or SDE solvers (Lu et al., 2022; Liu et al., 2022), direct mapping of noise to data (Song et al., 2023), distillation (Meng et al., 2023; Huang et al., 2024), enabling parallel sampling (Li et al., 2024; Wang et al., 2024; Chen

et al., 2024; Shih et al., 2024), and optimizing the inference of neural networks using methods like pruning (Fang et al., 2023), quantization (Li et al., 2023b), and reusing intermediate features (Ma et al., 2024; So et al., 2023; Wimbauer et al., 2024).

During the image generation process, diffusion models produce images at resolutions specified by user-defined parameters. However, there are instances when the generated image contains elements that may be irrelevant to the original user prompt. This observation led us to a key insight, motivating the development of a novel approach called **Highlight Diffusion**—a training-free method that reduces computational cost and latency at each step. Highlight Diffusion strategically prioritizes computation in regions that are most relevant to the given prompt, identified by the model during intermediate steps, while coarsely computing less significant regions. By focusing computational resources on these critical areas, our approach significantly reduces overall computation per step, thereby improving the efficiency of the diffusion process without sacrificing image quality.

To achieve this, we concentrated on two critical aspects of diffusion models. First, **feature redundancy** is an inherent characteristic due to the iterative denoising process, where diffusion models often exhibit significant similarities in intermediate feature maps across consecutive steps. These redundant features can be stored and reused for non-highlighted regions, eliminating the need for repeated computations in these areas.

Second, **cross-attention maps** are generated in modern text-to-image diffusion models that incorporate transformer architectures to guide image generation in accordance with the provided text prompt. The cross-attention mechanism fuses textual information with intermediate noise, generating a cross-attention map for each token. These maps capture semantic information of each token which we validated through extensive experiments. The cross-attention maps, which contain critical information about each token, are captured during intermediate steps and are used to identify and highlight regions that require fine-grained computation.

Through this method, **Highlight Diffusion** effectively reduces computational overhead while maintaining high-quality image outputs, addressing a key bottleneck in the practical application of diffusion models.

## 2 RELATED WORKS

Diffusion models have demonstrated significant versatility in a wide range of generative tasks, including text-to-image synthesis (Rombach et al., 2022; Podell et al., 2023; Saharia et al., 2022), image editing & inpainting (Meng et al., 2021; Tumanyan et al., 2023; Kawar et al., 2023), object detection (Chen et al., 2023), code generation (Singh et al., 2023), text generation (Li et al., 2022; Gong et al., 2022), and even audio generation (Schneider, 2023). Their ability to capture complex data distributions and produce high-quality outputs has made them a powerful tool across various domains. However, despite their generative prowess, diffusion models often face challenges related to computational efficiency, particularly due to the iterative nature of their sampling process. Consequently, several recent works have focused on accelerating diffusion models by leveraging architectural properties and feature redundancy.

Several recent works have explored methods to accelerate diffusion models by leveraging architectural properties and feature redundancy. *DeepCache* (Ma et al., 2024) capitalizes on the structure of U-Net and the redundancy present in intermediate feature maps to speed up the reverse diffusion process. The U-Net architecture consists of two main components: the skip branch and the main branch. The main branch which responsible for the majority of the computation, processes the full feature map. By caching and reusing the feature map from the previous timestep, DeepCache bypasses the costly computations in the main branch and performs only the lighter operations in the skip branch, resulting in significantly improved latency.

In text-guided diffusion models, the cross-attention layer within the transformer block integrates text embeddings with the noisy image $x_t$ at each timestep $t$, assigning weights to each pixel based on how much the prompt influences the image. *Prompt-to-Prompt Image Editing* (Hertz et al., 2022) demonstrates how the cross-attention layer can manipulate the spatial layout of an image according to the semantics of each word in the prompt. By leveraging this property, the method enables text-based image editing by selectively controlling which parts of the image are modified.

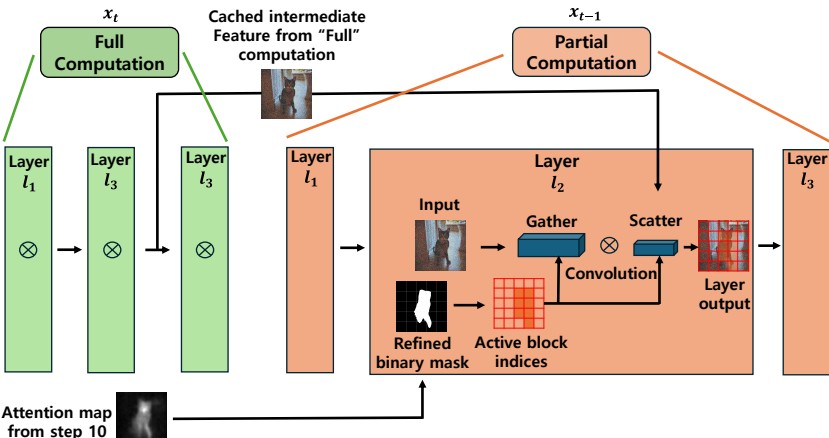

Figure 2: The intermediate feature map obtained from the previous "Full" computation, along with the refined binary mask generated at step 10, are pre-cached. Using this binary mask, the active block indices are identified. These active indices determine which blocks from the input feature map are gathered and stacked along the batch dimension. Subsequently, convolution operations are performed only on these selected blocks. The resulting blocks are then scattered back to the intermediate feature map from the previous "Full" computation to produce the final partially computed output.

Another notable work is *SIGE* (Li et al., 2023a), proposed by Li et al., which also exploits feature redundancy for image editing tasks. SIGE utilizes the *SBnet* (Ren et al., 2018) gather-and-scatter algorithm, which efficiently handles sparse convolutions to reduce computational overhead. Specifically, SIGE enhances image inpainting and editing by focusing computation only on the regions defined by the user edits. It gathers blocks of the intermediate feature map corresponding to the edited regions and passes them through the convolution and attention layers, thereby reducing the overall computational cost. The processed blocks are then scattered back into the original feature map, while the unedited regions reuse their previous feature values, maintaining efficiency and ensuring high-quality results.

## 3 METHODOLOGY

Our approach builds on unique characteristics of the diffusion model, particularly the observation of temporal redundancy in intermediate feature maps across consecutive steps. As demonstrated in Figure 2 we leverage this redundancy by reusing intermediate features from previous steps. To further enhance image quality, instead of reusing the entire set of intermediate features, we selectively recompute only the highlighted regions, identified using cross-attention maps generated at an intermediate step. This selective computation allows our method to reduce latency while maintaining high image quality, thus offering a more efficient generation process.

### 3.1 PRELIMINARY

Diffusion models are a class of generative models that create images by iteratively removing noise from a random sample, $\mathbf{x}_T$, which is drawn from an isotropic Gaussian distribution. The generation process consists of two primary phases:

**Forward Process.** In the forward process, the model progressively adds Gaussian noise to an image, effectively transforming it into pure noise over several steps. Mathematically, this can be described as a sequence of transformations applied to a data point $\mathbf{x}_0$. At each step $t$, noise is added according to a fixed variance schedule, resulting in a noisy version of the original image $\mathbf{x}_t$. The forward process can be summarized as:

$$q(\mathbf{x}_t|\mathbf{x}_{t-1}) = \mathcal{N}(\mathbf{x}_t; \sqrt{1-\beta_t}\mathbf{x}_{t-1}, \beta_t\mathbf{I})$$

where $\beta_t$ represents a variance term at each step, and $\mathcal{N}$ denotes the normal distribution.

**Reverse Process.** The reverse process aims to generate a new image by denoising $\mathbf{x}_T$ iteratively, starting from the final noisy sample $\mathbf{x}_T$ (which is a sample from a Gaussian distribution) back to $\mathbf{x}_0$. At each step, the model predicts the noise component and subtracts it to refine the image. The reverse process is parameterized by a neural network $\boldsymbol{\theta}$ and can be expressed as:

$$p_{\boldsymbol{\theta}}(\mathbf{x}_{t-1}|\mathbf{x}_t) = \mathcal{N}(\mathbf{x}_{t-1}; \boldsymbol{\mu}_{\boldsymbol{\theta}}(\mathbf{x}_t, t), \sigma_t^2\mathbf{I})$$

where $\boldsymbol{\mu}_{\boldsymbol{\theta}}$ is the mean function predicted by the neural network, and $\sigma_t^2$ is the variance at step $t$.

The iterative denoising procedure enables diffusion models to generate high-quality images by effectively learning the reverse of the noisy data transformation.

## 3.2 FEATURE REDUNDANCY

In modern diffusion models, the intermediate feature map $x_t$ at a given timestep $t$ often exhibits redundancy with the feature map $x_{t-1}$ from the consecutive timestep $t-1$. Similarly, we exploit this property by reusing parts of the intermediate feature maps that are deemed less important. The cross-attention map is used to identify the critical regions that require computation, while unimportant regions simply reuse feature map values from previous timesteps.

However, for every given interval $N$, the entire feature map is recomputed to capture global features, ensuring that the image quality remains high. This balance between selective computation and full recomputation enables efficient yet high-quality image synthesis.

## 3.3 CROSS ATTENTION MAPS

Cross-attention is an essential component of text-guided diffusion models, enabling the model to combine the input noisy image $x_t$ and the corresponding text embedding $\tau_c$. During the reverse diffusion process, the model's U-shaped network predicts the noise component $\epsilon$ based on these inputs. The cross-attention layer fuses this information by producing attention maps for each token in the text prompt, guiding the image generation process.

More specifically, the cross-attention layer projects the noisy image $x_t$ and the text embedding $\tau_c$ through learned weight matrices $l_q$, $l_k$, and $l_v$. This process generates the Query, Key, and Value representations, defined as:

$$Q = l_q(x_t)$$
$$K = l_k(\tau_c)$$
$$V = l_v(\tau_c)$$

The cross-attention map is then computed as:

$$M = \text{Softmax}\left(\frac{QK^T}{\sqrt{d}}\right) \tag{1}$$

where $M$ represents the attention map, and $d$ is the dimension of the key vectors. This map quantifies the relevance of each token in the text prompt to various image regions.

**Collecting Cross-Attention Maps.** We configure Highlight Diffusion to collect cross-attention maps after 20% of the total denoising steps for a good balance between the stability of the attention maps and reduced latency. We identify that the cross-attention mechanism in a diffusion model can be broadly divided into two stages. During the first stage, cross-attention primarily plans the overall structure of each token, starting at the initial denoising steps. This structure is consistently maintained throughout the entire denoising process. In the second stage, cross-attention gradually refines fine-grained details, aligning the attention map with the final image to be generated.

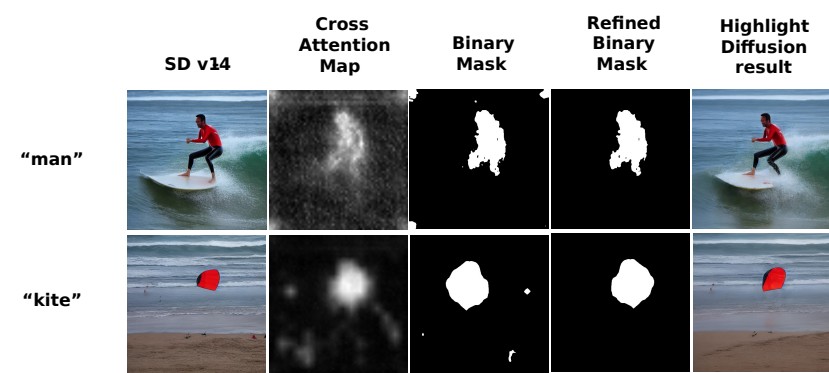

Figure 3: This figure demonstrates how the binary masks generated from cross-attention maps can sometimes contain white noise, which is irrelevant to the token being targeted for computation. The refining process eliminates this noise, reducing unnecessary computation and enabling the model to focus more effectively on the highlighted object.

In our model, we focus on capturing the cross-attention map during the first stage, when the overall structure is being planned. Due to the requirement of performing full computation before collecting cross-attention maps, capturing them at later stages can lead to increased latency, even though it may improve image quality. Our primary objective is to accelerate the image generation process. Capturing attention maps during the structure-planning stage proved sufficient for roughly identifying regions that could be partially computed. Please refer to the appendix for further details.

**Token Selection via Attention Map.** In order to select the most important tokens and their corresponding attention maps, we conducted an experiment using the *PartiPrompt* dataset, focusing on the animal category. Initially, we applied **Part-of-Speech (POS)** tagging to extract nouns from the given prompts, as nouns are typically the most relevant to visual content generation.

For each noun token, we computed the following statistics based on its associated attention map:

1. The sum of all values in the attention map.
2. The maximum value in the attention map.
3. The variance of the attention map.

We conducted this experiment across three different categories. Our results indicated that the token with the highest variance in its attention map most closely corresponded to the animal mentioned in the prompt, as classified under the *PartiPrompt* animal category data.

### 3.4 REFINING BINARY MASK

After selecting a critical token and its corresponding attention map, we generated a binary mask using a threshold value $h_{th} \in \{0, 255\}$. For this experiment, we set $h_{th} := 100$. However, this method often results in small white regions as shown in Figure 3 that are considered negligible within the binary mask, which causes inefficiencies during computation. Specifically, the gather-and-scatter algorithm assigns block indices to these small negligible regions, unnecessarily increasing the computational cost.

To enhance the accuracy of the binary mask, we employed the `cv2.connectedComponents` function to filter out regions with an area smaller than 1000 pixels in a 512×512 resolution image. This threshold corresponds to 0.4% of the total image area and is considered negligible in contributing meaningfully to the detected objects within the mask.

### 3.5 PARTIAL COMPUTATION WITH SIGE

To partially compute the highlighted regions identified by the refined binary mask, we incorporated the SIGE method proposed by (Li et al., 2023a). The identified regions, as determined by the refined

---

**Algorithm 1** Accelerated Diffusion Process Using Highlight Diffusion

---

**Parameters:**
    $x_t$: Current sample at time step $t$
    $t$: time step
    $h_c$: Conditioning information (e.g., class labels or context)
    $\epsilon_t$: Estimated noise at time step $t$
    $M$: Cross attention map computed by the full model
    $B$: Refined binary mask derived from $M$
    $N$: Interval for executing the full model computation
    $F_t$: cached intermediate feature from previous full model computation
    Full_Model: Original Diffusion Process (convolution performed on entire resolution)
    Partial_model: Highlight Diffusion Process (partial computation only on the highlighted region identified by the Binary_mask $B$

    Initialize $x_T \sim \mathcal{N}(0, I)$          ▷ Sample $x_t$ from normal distribution
    **for** $t = T, T-1, \ldots, T-9$ **do**
        $(\epsilon_t, M) = \text{Full\_Model}(x_t, h_c, t)$
        $x_{t-n} = \sqrt{\alpha_t}x_t - \sqrt{1 - \bar{\alpha}_t}\epsilon_t + \sigma_t z$
        **if** $t == T-9$ **then**
            $B = \text{Binary\_mask}(M)$          ▷ Process $M$ into binary mask $B$
            Store $F_t$ in cache
        **end if**
    **end for**

    **for** $t = T-10, T-11, \ldots, 1$ **do**
        **if** $t \mod N == 0$ **then**
            $\epsilon_t, M \leftarrow \text{Full\_Model}(x_t, h_c, t)$
            Store $F_t$ in cache
        **else**
            $\epsilon_t \leftarrow \text{Partial\_Model}(x_t, h_c, t, B, F_t)$
        **end if**
        $x_{t-n} = \sqrt{\alpha_t}x_t - \sqrt{1 - \bar{\alpha}_t}\epsilon_t + \sigma_t z$
    **end for**
**Output:** Final sample $x_0$          ▷ Output the final denoised sample

---

binary mask, are assigned active indices and grouped into equal-sized blocks. These blocks are gathered along the batch dimension and sub passed through the convolution layers. The computed blocks are then scattered back into the intermediate features that were cached during the previous full computation step. This process is similarly applied to the attention layers and is repeated for all convolution and attention layers within each module of the U-Net architecture. For more details, please refer to the original paper.

However, performing partial computation for highlighted regions throughout the entire denoising process results in significantly degraded image quality. To address this, Highlight Diffusion performs full computation at regular intervals, denoted as $N$, where the entire resolution of the latent variable $x_t$ is computed as in the original Stable Diffusion model. This strategy allows the model to integrate information from both highlighted and non-highlighted regions, ensuring the preservation of global semantics and ultimately generating higher quality images.

## 4 EXPERIMENT

### 4.1 EXPERIMENT SETTINGS

The experiment was conducted on NVIDIA GeForce RTX 3090 GPUs. We build upon the pre-trained Stable Diffusion V1.4 and the weights were acquired from repositories that were opened to the public. MS-COCO 2014 5k validation set (Lin et al., 2014) was used as prompts for both Stable Diffusion and Highlight Diffusion. Following previous works, we evaluated our metrics using

Table 1: Quantitative Evaluation of Highlight Diffusion compared to Stable Diffusion V1.4

| Model | Interval | Threshold | Latency (s) | Speedup ↑ (Averaged) | FID ↓ | CLIP ↑ | PSNR ↑ | LPIPS ↓ |
|---|---|---|---|---|---|---|---|---|
| **DDIM (50)** | - | - | **7.93** | **1.00** × | 27.67 | 31.54 | 8.74 | 0.84 |
| **HLDiffusion +DDIM (50)** | 2 | 80 | 6.60 | 1.20 × | 27.76 | 31.54 | 8.82 | 0.85 |
| | | 100 | 6.57 | 1.21 × | 27.86 | 31.53 | 8.82 | 0.85 |
| | | 120 | 6.26 | 1.27 × | 27.91 | 31.53 | 8.82 | 0.85 |
| | | 150 | 6.23 | 1.27 × | 27.83 | 31.54 | 8.82 | 0.85 |
| | 5 | 80 | 5.92 | 1.34 × | 28.03 | 31.53 | 9.01 | 0.86 |
| | | 100 | 5.81 | 1.36 × | 28.14 | 31.53 | 9.01 | 0.86 |
| | | 120 | 5.40 | 1.47 × | 28.23 | 31.53 | 9.01 | 0.87 |
| | | 150 | **5.22** | **1.52** × | 28.32 | 31.52 | 9.00 | 0.87 |
| | 10 | 80 | 5.77 | 1.37 × | 32.13 | 31.28 | 9.21 | 0.87 |
| | | 100 | 5.64 | 1.41 × | 31.32 | 31.32 | 9.21 | 0.87 |
| | | 120 | 5.16 | 1.41 × | 30.88 | 31.33 | 9.20 | 0.88 |
| | | 150 | 4.85 | 1.51 × | 30.18 | 31.37 | 9.18 | 0.88 |

Fretchet Inception Distance (FID) (Heusel et al., 2017), PSNR, LPIPS (Zhang et al., 2018), and CLIP Score (Hessel et al., 2021). we used clean-FID (Parmar et al., 2022) to calculate FID.

## 4.2 BASELINES

To the best of our knowledge, no existing research has addressed the partial computation of specific regions in text-to-image generation tasks. For example, the SIGE engine proposed by Li et al. was employed exclusively for image inpainting and image-to-image translation, which fundamentally differs from our approach. Consequently, we establish our baseline using DDIM with a number of score function evaluations (NFE) of 50 in Stable Diffusion V1.4 for comparison with our model.

## 4.3 FEATURE REDUNDANCY ANALYSIS

To demonstrate feature redundancy, we compared the resulting intermediate features from each layer in the U-Net architecture across consecutive time steps. We measured the cosine similarity of each feature map after flattening it into a one-dimensional tensor. For a total of 50 denoising diffusion implicit model (DDIM) (Song et al., 2020) steps, most of the changes occurred before step 10 and at the very last step of the denoising process. Based on these experimental results, we concluded that performing full computation for the first 10 steps is ideal for high-quality image generation. After step 10, it is reasonable to cache and reuse the intermediate features, given the minimal changes observed. Additionally, we set the interval such that it is a divisor of the total steps our our denoising process, ensuring that full computation is performed at the last step, which also showed a noticeable difference in our experiments. Please refer to the appendix for further details.

## 4.4 EXPERIMENTAL RESULTS

We demonstrate the effectiveness of our approach by evaluating the average latency on the MS-COCO 2014 validation set. Each prompt in this dataset generates a unique cross-attention map, which leads to a transformed and refined binary mask with a varying percentage of highlighted regions. Consequently, the generation time of our model differs depending on the specific prompt. To quantify this, we measure the average latency for generating all 5,000 images in the MS-COCO 2014 validation set and compare it with the time required by the original model with DDIM of 50 steps to generate the same set of images. We conducted experiments to give both quantitative and qualitative analysis of our model.

**Quantitative Analysis** As shown in Table 1, we measured the relative speed-up of our model compared to the baseline Stable Diffusion 1.4 in terms of latency, Fréchet Inception Distance (FID), and CLIP score. For both models, we utilized DDIM with 50 steps. As indicated in the table, our model achieved up to 1.52× speed-up, with a minor increase of 0.65 in FID and a negligible difference

| Methods | | Speedup | ΔFID ↓ |
|---|---|---|---|
| FRDiff (So et al. (2023)) | | 1.40× | 0.61 |
| **Highlight Diffusion (Ours)** | Interval:5 Threshold: 150 | **1.52×** | **0.65** |
| | Interval:5 Threshold: 120 | 1.47× | 0.56 |

Table 2: Acceleration speedups compared to a prior work

of 0.02 in CLIP score. When increasing the interval to 10, the FID score started to degrade significantly, without a corresponding improvement in latency. From these results, we infer that our model can generate images with minimal quality degradation up to an interval of 5. Additionally, increasing the binary mask threshold reduces the area for partial computation, resulting in faster sampling speeds but at the cost of decreased image quality.

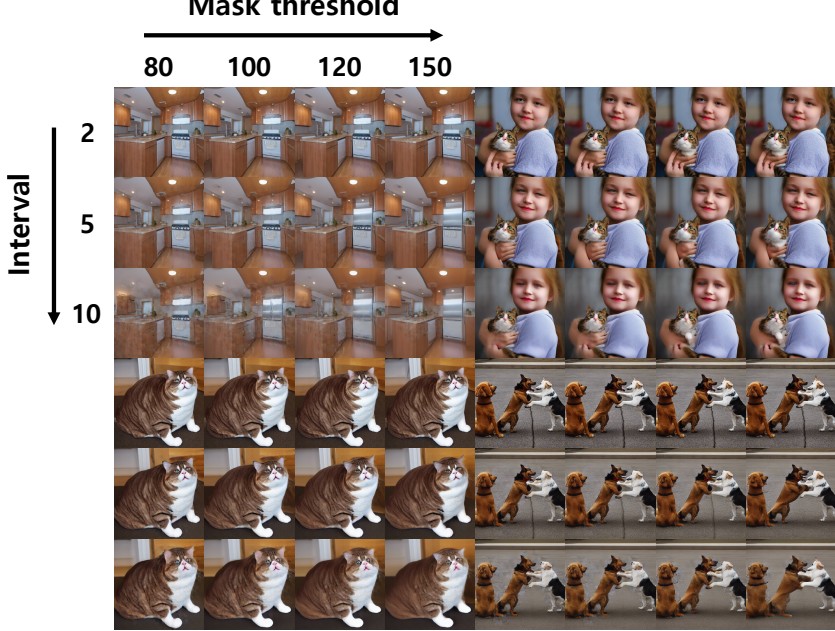

Figure 4: Visualization of different intervals and mask thresholds

**Qualitative Analysis** As shown in Figure 4, we compare the generated images from both Stable Diffusion V1.4 and HighLight Diffusion across different intervals. When the interval was set to 10, there was a significant degradation in image quality. The images appeared blurry and failed to capture high-frequency details of the objects. Additionally, visible boundaries emerged between the highlighted regions and the rest of the image due to lack of full computation. From both quantitative and qualitative perspectives, our results indicate that HighLight Diffusion can generate high-quality images up to an interval of 5.

**Comparison to Prior Works** We compare the relative speedups using existing diffusion acceleration methods. On our selected reference point (interval of 5 and mask threshold of 150), we observe FID score degradation of 0.65. As seen in Table 2, similar degradation level in FRDiff (So et al., 2023) shows a speedup of 1.40× while our work outperforms with a speedup of 1.52×.

## 5 LIMITATIONS

Our work has several limitations that merit consideration. First, the proposed method is specifically tailored for diffusion models that rely on text prompts for image generation. As a result, models that do not utilize text prompts are incompatible with our approach because they do not produce cross-attention maps—a core component of our technique.

Moreover, our proposed method shows limited merits in instances where a highlighted token corresponds to the global structure of the entire image. For example, when the input prompt is "a kitchen," our model might identify "kitchen" as the primary token, representing the overall image structure. In such cases, the corresponding cross-attention map's spatial structure is spread across the entire resolution, leading to similar latency to vanilla diffusion pipeline through DDIM. Future works may include developing methods that polarizes cross-attention maps to evaluate redundant computation to reduce latency even in these cases.

## 6 CONCLUSION

In this paper, we introduce HighLight Diffusion, a novel, training-free acceleration technique for text-to-image diffusion models. Our method leverages the unique characteristics of diffusion models, such as feature redundancy and the cross-attention mechanism. HighLight Diffusion utilizes cross-attention maps to identify and highlight significant regions, which are then partially computed, while reusing cached features for non-highlighted regions. Our research primarily focuses on reducing latency for individual time steps. This approach achieves a speed-up of up to $1.52\times$ compared to the original Stable Diffusion V1.4 with DDIM in the image generation process, with minimal degradation in image quality. We believe that our work opens new possibilities for accelerating diffusion models and advancing research in this area.

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

# A APPENDIX

## A.1 EXPLORING FEATURE REDUNDANCY

For this experiment, we used Stable Diffusion V1.4 with DDIM 50 steps. In our approach, we extracted the intermediate features computed by each `TimestepEmbedSequential` module within the U-Net at each timestep. We then compared these intermediate features at timestep $t$ with the corresponding features from the subsequent timestep $t - 1$. To quantify the similarity, we measured the cosine similarity between these intermediate features and plotted the results, as illustrated in Fig. 5. In this figure, each graph corresponds to a different `TimestepEmbedSequential` module, while the bottom-right graph integrates all the results to provide an aggregated view of the cosine similarity across layers and timesteps. Our results indicate that there is a significant change in the intermediate features for all layers before step 10 and at the very last step of the denoising process. This finding led us to perform full computation for the first 10 steps and then utilize cached features to perform partial computation for the remaining steps within the specified intervals parameter.

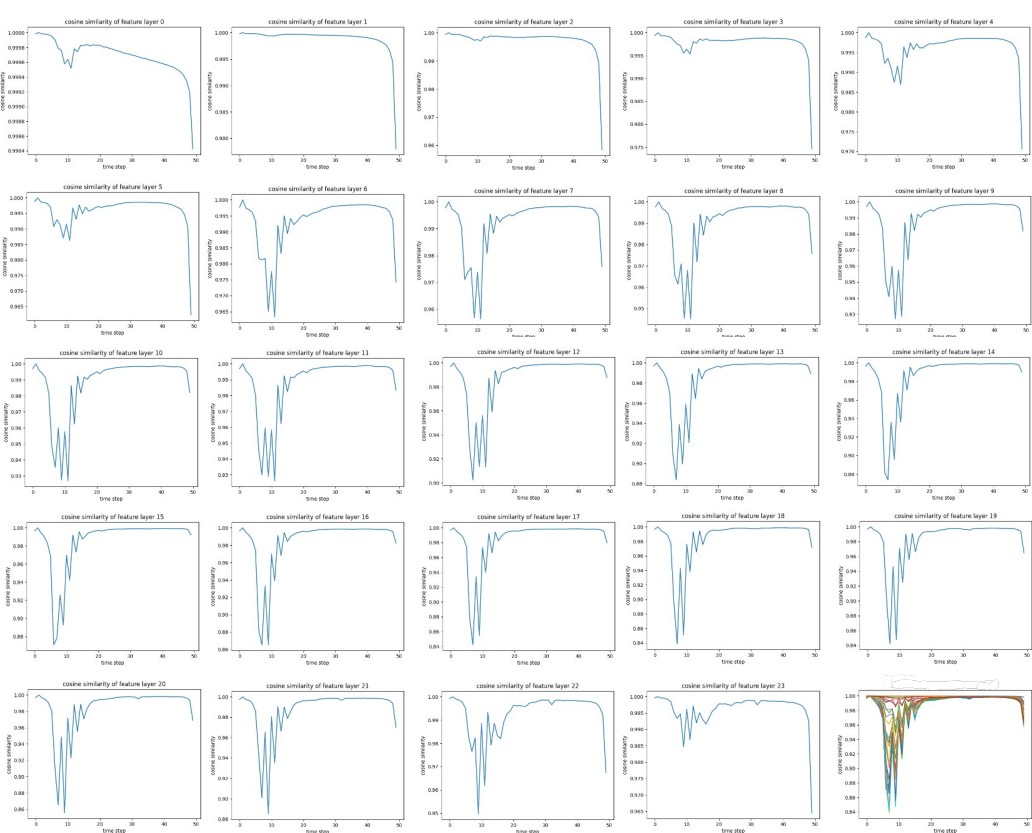

Figure 5: Shows the cosine similarity of intermediate features of each layer. As exhibited, the difference between feature maps are significant before step 10 and at the very last step.

## A.2 EXPLORING BINARY MASK GENERATION FROM CROSS-ATTENTION MAPS

In this section, we explore additional methods (choosing sum and maximum values as reference for attention token) for generating binary masks from cross-attention maps. The results of various mask generation techniques are visualized in Figure 6. Our experiments reveal that using the sum or maximum values as the attention token reference often produces suboptimal attention masks that fail to adequately cover critical regions. We hypothesize that this limitation contributes to the degradation in generation quality, as supported by the experimental results presented in Table 3.

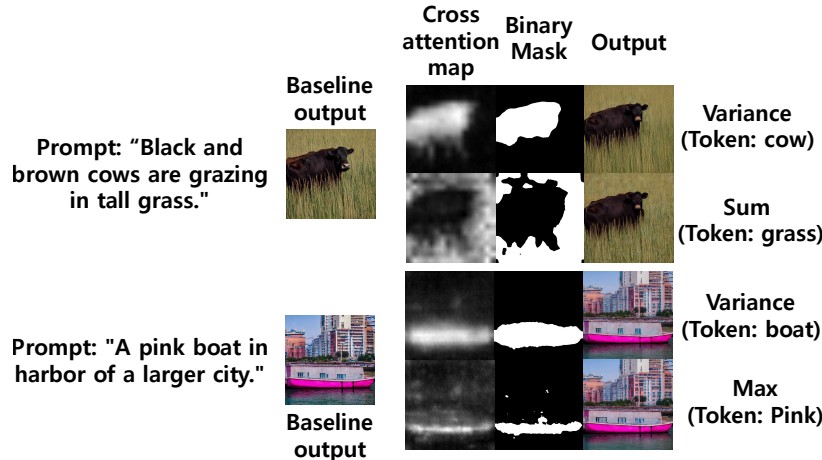

Figure 6: Visualization of Different intervals and mask thresholds

Table 3: Effect of Mask Generation Methods

| Model | Interval | Threshold | Speed (s) | Speedup ↓ | FID ↓ | CLIP ↑ |
|---|---|---|---|---|---|---|
| SDv1.4 | - | - | 7.93 | 1.00 × | 27.67 | 31.54 |
| HLDiffusion (var) | 5 | 100 | 5.81 | 1.36 × | 28.14 | 31.53 |
| HLDiffusion (max) | 5 | 100 | 5.73 | 1.41 × | 29.36 | 31.53 |
| HLDiffusion (sum) | 5 | 100 | 5.97 | 1.47 × | 28.98 | 31.53 |

### A.3 ANALYZING THE RELATIONSHIP BETWEEN MASK SIZE AND LATENCY

To better understand the correlation between mask size in Highlight Diffusion and the resulting speedup, we examine the relationship between the total number of blocks gathered from binary masks and the average latency per step, as shown in Figure 7. While the theoretical expectation suggests a strictly proportional relationship—given that the number of multiply-accumulate operations (MACs) is proportional to the number of blocks—our observations reveal a minimum latency of 30ms, even for a block size of 1. This suggests diminishing returns when using very small masks, indicating that there is a lower bound to the latency, beyond which further mask reduction offers limited speed improvements. Future work may investigate the source of this limitation, identifying potential bottlenecks that constrain the efficiency gains at smaller mask sizes.

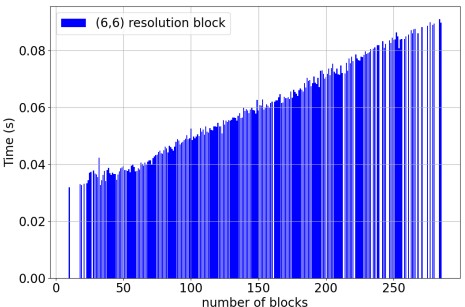

Figure 7: Per-step latency plotted against the number of blocks gathered from the binary masks during the diffusion process.

### A.4 SAMPLES

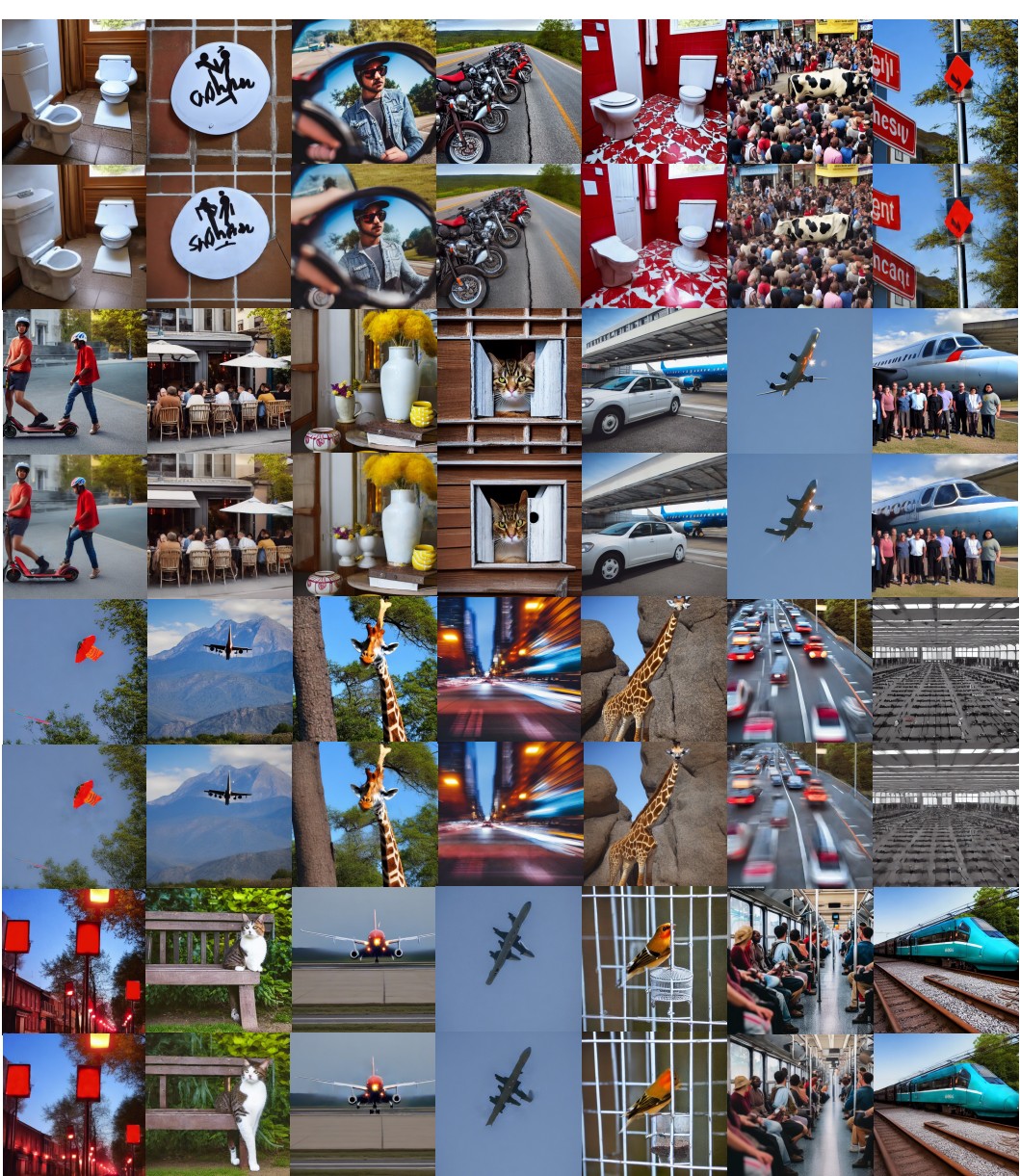

Figure 8: DDIM 50 steps for MS-COCO.Samples from Baseline (upper line). Samples from High-Light Diffusion (lower line)

