# OpenReview forum: "Highlight Diffusion: Training-Free Attention Guided Acceleration for Text-to-Image Models"
_ICLR.cc/2025/Conference — ICLR 2025 Conference Withdrawn Submission_

### Official Review · Reviewer_fXBJ · 2024-10-23

**Soundness:** 3
**Presentation:** 3
**Contribution:** 2
**Rating:** 3
**Confidence:** 4

**Summary:**

The paper introduces Highlight Diffusion, a training-free method that accelerates text-to-image diffusion models by using cross-attention maps to focus on critical image segments during the denoising process. This approach achieves a 1.52× speedup with minimal quality loss, indicated by slight increases in FID and decreases in CLIP scores, providing an efficient solution for practical deployment of diffusion models.

**Strengths:**

1. This paper addresses an important issue: the acceleration of diffusion models. The authors propose a selective mechanism that automatically identifies crucial parts of the generated image via cross-attention maps and emphasizes denoising in these regions. This presents an intriguing solution to improving efficiency in diffusion models.

2. The paper is well-written and easy to follow.

**Weaknesses:**

1. Limited Technical Contribution: The primary idea of the paper is to selectively compute highlighted regions during the denoising process, which allows for significant reductions in computational demands. However, if the object mentioned in the prompt is large and occupies a significant portion of the image, the proposed method may struggle to reduce computation effectively. In such cases, the approach appears to resemble the previous work, DeepCache [1]. The authors should clarify the distinctions between their method and DeepCache.

2. Handling Text-Image Misalignment: Text-to-image models often experience text-image misalignment. How does the proposed method handle such misalignments, and what strategies are in place to ensure consistent performance in the presence of these issues?

3. Insufficient Experiments: The experimental section includes too few baselines, with only one competing method, FRDiff. The authors should compare their approach against a broader range of acceleration methods, such as DeepCache [1], Diff-Pruning [2], SnapFusion [3], and Spectral Diffusion [4], to provide a more comprehensive evaluation.

4. Generalization Issues: The authors only conduct experiments on SD1.4, which raises concerns about the generalization of the proposed method. Additionally, SD1.4 is somewhat outdated. The authors should validate their approach on other more recent diffusion models, such as SDXL, SD3, and Flux, to demonstrate its applicability across different architectures.

[1] DeepCache: Accelerating Diffusion Models for Free. CVPR 2024.

[2] Structural pruning for diffusion models. NeurIPS 2023.

[3] Snapfusion: Text-to-image diffusion model on mobile devices within two seconds. Arxiv 2023.

[4] Diffusion probabilistic model made slim. NeurIPS 2023.

**Questions:**

Please refer to the weakness section.

---

### Official Review · Reviewer_SfZa · 2024-10-23

**Soundness:** 3
**Presentation:** 2
**Contribution:** 2
**Rating:** 3
**Confidence:** 4

**Summary:**

The authors introduce Highlight Diffusion, a novel training-free acceleration approach. It achieves a noticeable speedup while maintaining generation quality via an attention-guided generation process. Specifically, Highlight Diffusion strategically prioritizes computation in regions most relevant to the given prompt, which are identified by the model during intermediate steps, while roughly computing less significant regions. Through this method, it effectively reduces computational overhead and maintains high-quality image outputs. As a result, Highlight Diffusion achieved a 1.52× speedup using an NVIDIA RTX 3090 GPU without significant image quality degradation.

**Strengths:**

The method is presented clearly. The code in the supplementary materials verifies the details, enhancing the method's credibility and facilitating understanding and potential reproduction.

**Weaknesses:**

1. The overall structure of the draft could be improved. Section 3.1 seems unnecessary as it may not contribute significantly to the core understanding of the proposed method. Some content in the appendix appears to be more relevant and could have been incorporated into the main paper. Additionally, there are a lot of empty spaces in the main paper, which gives an impression of poor space utilization and may indicate that the work could have been better refined.

2. While the achieved 1.52× speedup is commendable, the application scope of this method is rather narrow. There are only a few works where it can be effectively applied. For example, as seen with the “cat” example (line 798), it does not seem to work in certain cases. This limited applicability restricts the practical value and generalizability of the method.

3. The experiment appears to be insufficient. In line 293, the setting of N is described as an empirical hyperparameter based on Table 1, but there is a lack of in-depth exploration. It is not clear whether it should always maintain the same value. There could have been more investigations into different settings such as using [N] = [1,3,5,7,9] or [9,7,5,3,1], etc. to determine if more efficient results could be obtained.

**Questions:**

Could you please provide more detailed explanations regarding the reasons behind the speedup achieved by Highlight Diffusion? Is it because certain tokens from the previous step do not participate in the computing process, or are there other factors at play? As it stands, this aspect is not clear to me. I would appreciate a more in-depth discussion on how exactly the method manages to accelerate the image generation process while maintaining the quality of the output. Additionally, it would be beneficial to understand if there are any trade-offs or limitations associated with this speedup mechanism that may not have been fully explored in the paper.

---

### Official Review · Reviewer_4exm · 2024-11-03

**Soundness:** 3
**Presentation:** 1
**Contribution:** 2
**Rating:** 3
**Confidence:** 4

**Summary:**

This paper proposes a training-free novel acceleration approach, called Highlight Diffusion, which achieves speedup while retaining comparable quality with baseline. By utilizing cross-attention maps to identify crucial segments, HLDiffusion can partially calculate the significant regions. With a minimal loss in generated image quality, Highlight Diffusion achieves a 1.52× speedup.

**Strengths:**

1. This paper explores feature redundancy in great detail in the appendix.
2. This paper refines the cross-attention mask, which is conducive to further research on the T2I diffusion model.

**Weaknesses:**

1. As a work to accelerate the T2I diffusion model, HLDiffusion needs to be compared with more Sota acceleration works, such as SpeedUpNet (ECCV 2024). There are too few comparisons in the experimental part.
2. SDM has developed to SD v1.5, SD XL, SD 2.0 and SD V2.1. And the most commonly used one is actually SD V1.5. Is the baseline selected in the experiment outdated?
3. Speed. A mere 1.52 x speedup does not seem convincing.
4. Writing quality. The core innovations of this paper are feature redundancy and refined masks, which occupy only a small part of the manuscript.
5. Typo. What is $x_{t-n}$ in the pseudocode of Algorithm 1?

**Questions:**

As mentioned above in the weaknesses, the most important thing is that you should show more comparison results of baselines and additional state-of-the-art methods.

---

### Official Review · Reviewer_5rH8 · 2024-11-04

**Soundness:** 2
**Presentation:** 1
**Contribution:** 2
**Rating:** 3
**Confidence:** 5

**Summary:**

The paper proposes a method to accelerate the inference of diffusion models without requiring additional training. The authors introduce a technique called Highlight Diffusion, which leverages cross-attention maps to focus computational resources on key areas relevant to the input text prompt. The authors report a 1.52× speedup on an NVIDIA RTX 3090 GPU, with minimal degradation in image quality, evidenced by slight changes in FID and CLIP scores.

**Strengths:**

+ The core idea is interesting. Leveraging salient regions in the attention map to select the most important tokens enables the model to focus on generating specific areas, thereby accelerating the process. The authors also introduced a caching mechanism to avoid redundant computations.
+ The authors validated the effectiveness of the algorithm on the MS-COCO dataset. It can be observed that the inference speed of the original model increased without significant loss in performance.

**Weaknesses:**

- The writing and organization of the paper are major issues. The writing is unclear, and many sentences are convoluted. Additionally, the authors dedicate too much space to introducing background knowledge, with very limited space allocated to experiments.
- The experiments do not sufficiently demonstrate the effectiveness of the algorithm, and its applicability is highly limited. Since the algorithm operates based on the most salient regions in the attention map, it is essentially object-centric. This might make it challenging to generalize to multi-object scenarios or cases focused purely on background or style image generation. I suggest that the authors test on benchmarks like DPG-Bench, which provide a more comprehensive evaluation of text-to-image generation capabilities, rather than simply using prompts from MS-COCO.
- The authors only validated the algorithm on SD1.4 and did not test its effectiveness on more popular or larger models (i.e., SDXL, Pixart-\alpha, FLUX). Additionally, it would be insightful to see how the algorithm performs when used in conjunction with adapters such as ControlNet or IP-Adapter.

**Questions:**

Please see the Weaknesses.

---

### Author Response · Authors · 2024-11-28
**Sincere Apologies**

I hope this message finds you well.
I am writing to express my sincerest apologies for the shortcomings in my paper submission, HighLight Diffusion.
I would like to also apologize, that I will not be able to conduct the additional experiments you requested within the given revision deadline.

Your feedback was invaluable in identifying critical areas for improvement, including:
1) Conducting comprehensive algorithm comparisons.
2) Testing on updated models, such as SDXL.
3) Incorporating experimental results using the DPG-Testbench.

I deeply regret that the time and resources required to perform these essential experiments exceed the constraints of the current revision schedule.
Please know that I am fully committed to addressing these issues in a thorough and systematic manner.
I sincerely apologize for any inconvenience or frustration caused by this situation.
I greatly value the time and effort you invested in reviewing my work.
I also truly thank you for providing constructive feedback, and I am determined to make the necessary improvements.

---

### Note · Authors · 2024-11-28

**Comment:**

I hope this message finds you well. I am writing to express my sincerest apologies for the shortcomings in my paper submission, HighLight Diffusion. I would like to also apologize, that I will not be able to conduct the additional experiments you requested within the given revision deadline.

Your feedback was invaluable in identifying critical areas for improvement, including:

Conducting comprehensive algorithm comparisons.
Testing on updated models, such as SDXL.
Incorporating experimental results using the DPG-Testbench.
I deeply regret that the time and resources required to perform these essential experiments exceed the constraints of the current revision schedule. Please know that I am fully committed to addressing these issues in a thorough and systematic manner. I sincerely apologize for any inconvenience or frustration caused by this situation. I greatly value the time and effort you invested in reviewing my work. I also truly thank you for providing constructive feedback, and I am determined to make the necessary improvements.

**Withdrawal Confirmation:**

I have read and agree with the venue's withdrawal policy on behalf of myself and my co-authors.